# Lightweight Vision Transformer Coarse-to-Fine Search via Latency Profiling

## Abstract

Despite their impressive performance on various tasks, vision transformers (ViTs) are heavy for mobile vision applications. Recent works have proposed combining the strengths of ViTs and convolutional neural networks (CNNs) to build lightweight networks. Still, these approaches rely on hand-designed architectures with a pre-determined number of parameters. This requires re-running the training process to obtain different lightweight ViTs under different resource constraints. In this work, we address the challenge of finding optimal lightweight ViTs given constraints on model size and computational cost using neural architecture search. Using the proposed method, we first train a supernet, which is a hybrid architecture of CNNs and transformers. To efficiently search for the optimal architecture, we use a search algorithm that considers both model parameters and on-device deployment latency. This method analyzes network properties, hardware memory access pattern, and degree of parallelism to directly and accurately estimate the network latency. To prevent the need for extensive testing during the search process, we use a lookup table based on a detailed breakdown of the speed of each component and operation, which can be reused to evaluate the whole latency of each search structure. Our approach leads to improved efficiency compared to testing the speed of the whole model during the search process. With extensive experiments on ImageNet, we demonstrate that under similar parameters and FLOPs, our searched lightweight ViTs have higher accuracy and lower latency compared to state-of-the-art models. For example, our AutoViT_XXS (71.3% Top-1 accuracy and 10.2ms latency) has a 2.3% higher accuracy and 4.5ms lower latency compared to MobileViT_XXS (69.0% Top-1 accuracy of and 14.7ms latency).

## 1 Introduction

Vision Transformer (ViT) Dosovitskiy et al. (2020) exploits the self-attention mechanism inherited from the transformer architecture and has recently obtained state-of-the-art performance in vision tasks like image classification Dosovitskiy et al. (2021); Touvron et al. (2021); Chen et al. (2021b), object detection Carion et al. (2020); Dai et al. (2021a); Amini et al. (2021); Misra et al. (2021); Dai et al. (2022), semantic segmentation Zheng et al. (2021); Cheng et al. (2021); Ding et al. (2021), image retrieval El-Nouby et al. (2021), image enhancement Yang et al. (2020); Chen et al. (2021c); Lu et al. (2021). These efforts have focused on improving the overall performance of the vision transformers. While the overall results are impressive, ViTs sacrifice lightweight model capacity and portability for high accuracy. Vision transformers are relatively less explored for mobile vision tasks Mehta & Rastegari (2021) and are difficult to deploy to edge devices due to resource constraints. The popular ViT models are made small by simply scaling them down Touvron et al. (2021) with a significant reduction in model performance. This challenge can be addressed by the careful manual design of the architecture to meet various resource constraints of different devices. However, this process requires several trials and errors to obtain viable candidates, especially for designing hybrid CNN-Transformer ViTs that have emerged recently. d'Ascoli et al. (2021); Chu et al. (2021); Dai et al. (2021b); Liu et al. (2021); Graham et al. (2021); Wang et al. (2021b).

While previous works have showcased improvements in on-device efficiency, they often do not prioritize consideration of the real hardware latency when designing the DNN models. Some contain mobile-unfriendly operations that limit performance improvement. LeViT Graham et al. (2021) employs a convolutional stem in place of the original patch stem. While LeViT successfully reduces FLOPs, it does so at the expense of introducing redundant parameters. This becomes a significant constraint for edge devices with limited memory capacity. Besides, relying on FLOPs as a performance

metric is misleading since it doesn't necessarily correlate with latency. Additionally, the reshaping process–a data layout change, can cost GPU/NPU cycles, which computation amounts can not be calculated. Also, HardSwish is not inherently supported by iPhone's CoreML. Sophisticated attention mechanisms, like the ones employed in the Long-Short Transformer Zhu et al. (2021), are notoriously hard to support or accelerate on mobile devices.

Neural architecture search (NAS) has shown its benefit over manual design as a powerful technique for the automatic design of neural networks, especially for designing efficient models. Autoformer Chen et al. (2021d) is a dedicated one-shot architecture search framework for pure transformer structures. However, their limited search space leads to performance degradation for very lightweight models. NASViT Anonymous (2022) searches for a conv and transformer hybrid structure. However, it has a prolonged training time. The gradient optimization method introduced to circumvent the conflicts between larger and smaller subnets is time-consuming. It also includes mobile-unfriendly operations such as the talking head module and window attention. S3 Chen et al. (2021e) proposes an automatic search space design method to improve the effectiveness of the design space. However, its evolutionary search is time-consuming. Our work endeavors to rectify the deficiencies found in existing solutions by introducing a truly efficient, hardware-oriented approach. This approach has been optimized to seamlessly adapt to the constraints of the target hardware and fulfill the specific speed requirements.

To address the existing ViT search problem, we propose an efficient NAS approach that considers three perspectives: optimal search space, training efficiency, and latency-guided search. Specifically, we design a new search space that enables the search for hybrid structures of convolution and transformer, which outperform existing hybrid structures, as demonstrated in Figure 1. To improve training efficiency, we encode the search space into independent supernets, which reduces optimization interference between subnets of different sizes. Our experiments show that this approach leads to improved performance compared to existing methods.

In the field of edge computing, the selection of models pivots critically on the metric of latency. Despite this, numerous prevailing search algorithms face challenges in accurately evaluating model latency using only FLOPs and parameter counts. As such, a model that appears smaller in size as a result of a search process does not necessarily guarantee higher speed. In light of these limitations, some researchers have made attempts to incorporate latency into the search criteria via the utilization of a

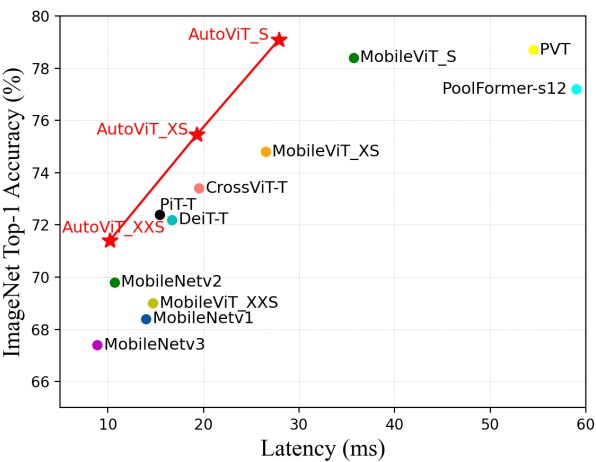

Figure 1: Top-1 accuracy and sizes of different models on ImageNet. Our method achieves a better trade-off than other efficient CNN-based and transformer-based models.

latency predictor Wang et al. (2020). However, developing an accurate predictor is not without its challenges—it necessitates hundreds, even thousands, of actual speed tests across diverse model structures to form a robust training dataset. If the search space is expansive, a proportional increase in actual test results is required to ensure accurate predictions across all possible architectures. To this end, we propose a coarse-to-fine search algorithm that utilizes latency as a primary indicator and a parameter as a memory threshold to explore the search space and find the optimal architecture. We leveraged the latency profiling technique to improve efficiency. As a result of our efforts, we successfully identified models with significantly lower latency than the previous approach of relying on flops and the number of parameters. Our contributions can be summarized as follows:

- **Hybrid Search Space:** We designed a search space that combines CNN and transformers, leveraging the strengths of both, to create lightweight ViTs suitable for mobile vision tasks.

- **Multi-Size Supernet Training Scheme:** To address the challenges of dealing with a large search space in vision transformers and optimize the subnets of different sizes, we proposed a multi-size supernet training scheme that mitigates optimization interference between subnets and enables efficient training.

- **Latency-aware Search Scheme:** We revisit the design principles of ViT and its variants through latency analysis and propose a latency profiling model, which can efficiently estimate the latency of network candidates.

We propose a parameter-latency-oriented coarse-to-fine search strategy to find the optimal subnets for hardware deployment among the trained supernets.

- **Performance:** Our searched model can achieve up to 79.2% accuracy on ImageNet with 6.3M parameters and 1.4 GFLOPs, exceeding the performance of existing models with similar resource budgets.

## 2 Related Work

**Vision Transformers.** ViT Dosovitskiy et al. (2020) is a pioneering work that uses only transformer blocks to solve various vision tasks. Compared to traditional CNN structures, ViT allows all the positions in an image to interact through transformer blocks. In contrast, CNNs operate on a fixed-sized window with restricted spatial interactions, which hinders their ability to capture relations at the pixel level in both spatial and time domains. Since then, many new variants have been proposed. For example, DeiT Touvron et al. (2021), T2T-ViT Yuan et al. (2021b), and Mixer Tolstikhin et al. (2021) tackle the data-inefficiency problem in ViT by training only with ImageNet. PiT Heo et al. (2021) replaces the uniform structure of the transformer with a depth-wise convolution pooling layer to reduce spacial dimension and increase channel dimension. SViTE Chen et al. (2021f) alleviates training memory bottleneck and improves inference efficiency by co-exploring input token and attention head sparsity.

**Neural Architecture Search.** There has been increasing interest in designing efficient networks with neural architecture search (NAS). Among different methods, weight-sharing NAS has become popular due to training efficiency Yu et al. (2020); Wang et al. (2021a); Sahni et al. (2021). They train one over-parameterized supernet whose weights are shared across all sub-networks in the search space to conduct architecture search, significantly reducing the computational cost. This one-shot NAS approach usually contains two phases. In the initial phase, all candidate networks within the search space are refined using weight sharing, ensuring that every network concurrently achieves optimal performance by the conclusion of training. The subsequent phase employs conventional search methods, like evolutionary algorithms, to identify the top-performing models, considering different resource limitations. BigNAS Yu et al. (2020) trains a supernet that covers all subnets in the architecture search space with weight sharing. AttentiveNAS Wang et al. (2021a) improve existing two-stage NAS with attentive sampling of networks on the best or the worst Pareto front.

**Efficient ViT Design.** There's been an increasing emphasis on developing efficient ViT through both NAS-driven designs and hand-crafted approaches. Each method has showcased efficiency gains, as evidenced by hardware evaluations. Regarding hand-crafted designs, LeViT Graham et al. (2021) replaces the original patch stem with a convolutional stem, enhancing inference speed for image classification. MOAT Yang et al. (2022) seamlessly integrates the advantages of mobile convolution and self-attention within a single block through a meticulous redesign. Regarding NAS-driven designs, HAT Wang et al. (2020) searches for an encoder-decoder transformer structure and requires additional retraining or finetuning of the optimal candidates obtained during the search. BossNAS Li et al. (2021a) searches for CNN-transformer hybrid models with block-wisely self-supervised neural architecture search. CvT Wu et al. (2021) proposes a new architecture family and searches for strides and kernel size. Autoformer Chen et al. (2021d) entangles the weights of different vision transformer blocks in the same layer during supernet training and trains three supernets with different scales to reduce training time. NASVIT Anonymous (2022) extends the search space to get efficient models leveraging a hybrid architecture of convolutions and transformers. The current designs and methods, however, have their set of limitations. These include the incorporation of operations that are not mobile-friendly, an over-reliance on parameters and FLOPs as primary evaluation metrics, restrictive search spaces, and prolonged training or search durations.

Our advantages over existing works can be summarized as follows: ① In contrast to hand-crafted methods, leveraging network search allows us to pinpoint the optimal structure tailored to specific hardware constraints. Hand-crafted designs tend to yield a set of models that primarily differ in their dimensions. While they may serve as general-purpose models suitable for a broad range of devices, they may not be as finely optimized for a specific hardware profile as network-searched models. We incorporated a hybrid search space with an inductive bias. This not only broadens the search space but also accelerating convergence and mitigating local optima. ② From a hardware-oriented perspective, it is more efficient to define a narrow search space tailored to specific constraints of target devices. As a result, there's no need for training with the entire search space. By segmenting the solution into multiple supernets, not only can we sidestep conflicts, but we can also dramatically cut down on training overhead. ③ Instead of the traditional approach of

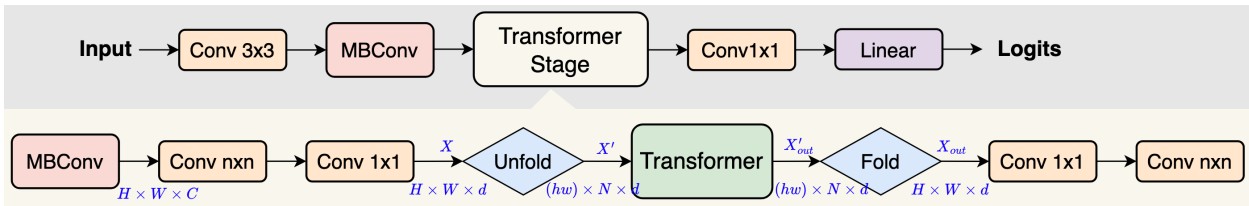

Figure 2: An illustration of our ViT structure. Conv and MBConv refer to standard convolution and inverted residual blocks, respectively. All CNN and transformer blocks contain a stack of dynamic layers with searchable architecture configurations. We also search for the input resolutions.

relying on hardware deployment for every candidate or using a latency predictor, our method stands out in its accuracy and efficiency. Our latency prediction model is a training-free theoretical model suitable for general-purpose hardware, GPU. It considers the properties of the target hardware, the model type, the model size, and the data granularity. It then quantitatively captures both the computation latency and data movement latency, enabling it to precisely predict the actual throughput for each layer.

## 3 Methodology

### 3.1 Background

Our goal is to search for lightweight and high-performance models for deployment on edge devices. There are two problems with current ViT models: (i) Most of them sacrifice efficiency to improve accuracy. The large computation cost and the number of model parameters make it difficult to deploy these models to devices such as cell phones or FPGAs. For example, Swin transformer Liu et al. (2021) achieves SOTA accuracy on multiple computer vision tasks. In the field of object detection, it is only applied to frameworks such as Retina Lin et al. (2017) or Maskrcnn He et al. (2017), but has not yet been applied to frameworks such as YOLO series Redmon et al. (2016); Redmon & Farhadi (2017; 2018); Bochkovskiy et al. (2020) that are known for their efficiency. (ii) Most of the current works are done by simply scaling down from the original dimension to obtain models of different sizes Dosovitskiy et al. (2021); Touvron et al. (2021). This coarse-grained model selection significantly sacrifices the accuracy of small models and offers limited flexibility in adjusting the size. Despite superior performance in the high computational budget regime, ViTs still do not perform as well as their CNN counterparts on small or medium-sized architectures, especially when compared to CNN architectures that are highly optimized by neural architecture search. CNN networks such as MobileNets Howard et al. (2017), ShuffleNetv2 Ma et al. (2018), ESPNetv2 Mehta et al. (2019), and MNASNet Tan et al. (2019) can easily replace the heavyweight backbones in existing task-specific models to reduce the model size and improve latency. One major drawback of these approaches is that they are spatially localized. On the other hand, the transformer is able to learn global representations, but ignores the spatial induction bias inherent in CNNs and thus requires more parameters to learn visual representations.

Based on these considerations, we focus on combining the advantages of CNN (e.g., spatial induction bias) and ViT (e.g., input adaptive weighting and global processing) to find a hybrid architecture of convolution and transformer that considers both performance and efficiency. Past work Mehta & Rastegari (2021) has shown that incorporating the downsampling mechanism of convolution into the transformer architecture can effectively reduce the size of the model and its ability to process high-resolution images, which can be of great benefit to the learning and deployment of the transformer. Due to the efficiency of local computation, convolution is introduced to process high-resolution inputs, while the transformer is used to process low-resolution features to extract global information.

### 3.2 Search Space

As shown in Figure 2, each of our hybrid blocks consists of standard convolution layers and one transformer block. Transformer stands for the transformer blocks and MBConv refers to MobileNetv2 Sandler et al. (2018) blocks. Standard ViT reshape input tensor $X$ of size $H \times W \times C$ into $N \times d$. Where $C$, $H$, and $W$ represent the channels, height, and width, $N$ represents the number of patches and d is the dimension. However, reshaping into 2D feature ignores the spatial inductive bias that is inherent in CNNs. Following Mehta & Rastegari (2021), we apply a $n \times n$ standard convolution layer followed by a point-wise convolution layer to produce $X$. The convolution layer encodes local

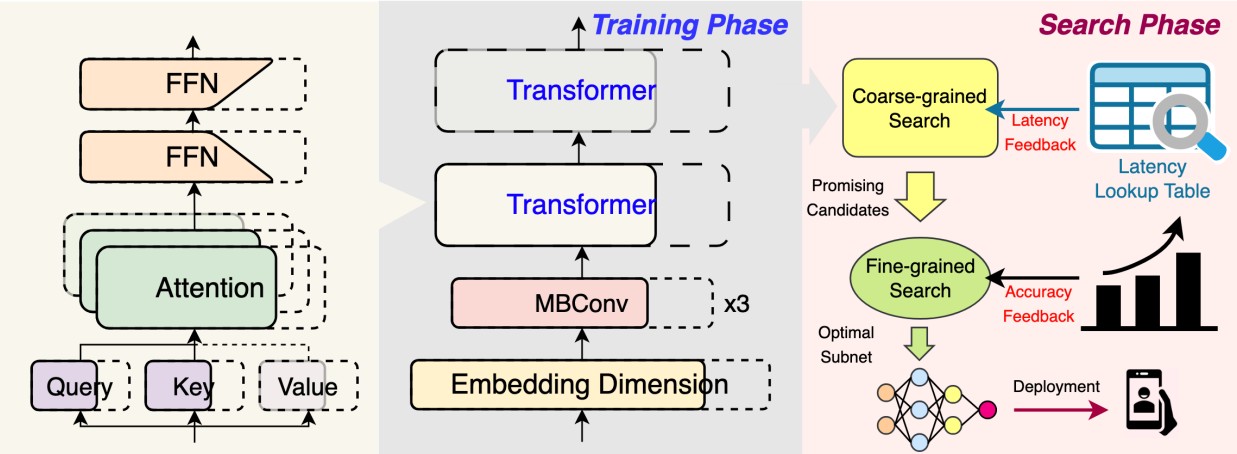

Figure 3: **Framework Overview.** We train a weight-shared supernet by iteratively optimizing randomly sampled subnets. *Left:* We search for the number of heads and expansion ratio of a transformer block. *Middle:* We search for the width and depth of MBConv and transformer block. Each layer and block are dynamic. Solid lines and dark blocks represent selected components in contrast to the dashed lines and lighter blocks. *Right:* Perform a coarse-to-fine search with hardware latency constraints to find the model with the highest validation accuracy.

spatial information while the point-wise convolution projects the tensor to a high-dimensional space by learning linear combinations of the input channels. To learn global representations with spatial inductive bias, we reshape (unfold) $X$ into $N$ non-overlapping flattened patches $X'$ of size $hw \times N \times d$, where $hw$ is the number of pixels of one patch $P$. The folding-unfolding process replaces local processing in convolutions with global processing using transformers. This allows the transformer block to have CNN- and ViT-like properties, which helps it learn better representations with fewer parameters and simple training recipes. For each patch, inter-patch relationships are encoded by applying transformers for each pixel to obtain $X'_{out}(p) = Transformer(X'(p))$, where $1 \le p \le P$. This type of transformer block prevents ViT from losing either the patch order or the spatial order of pixels within each patch. After that, we fold reshape (fold) $X'_{out}$ back to $X_{out}$ of size $H \times W \times C$. The spatial order of pixels within each patch will be retained throughout the process.

We summarize the detailed dimensions of our search space. In Table 1, width represents the channel size for CNN layers and hidden dimension for transformer layers, respectively. The depth denotes the number of repeated CNN and transformer layers for each block. The expansion ratio refers to the expansion ratio of the depth-wise convolution layer for CNN layers and the MLP expansion ratio for transformer layers. For each CNN block, we search for the optimal channel widths, block depths, expansion ratios, and kernel sizes. For each transformer block, we search for the best number of windows, hidden feature dimensions, depths, and MLP expansion ratios. Following one-shot NAS methods, we encode the search space into a supernet. All subnets share the weights of their common parts. The supernet is the largest model in the space. In particular, the supernet stacks the maximum number of transformer blocks with the largest embedding dimension, Q-K-V dimension, and MLP ratio as defined in the space. During training, all possible subnets are uniformly sampled, and the corresponding weights are updated.

Our transformer block structure, along with the search space allows us to search for lightweight models. This is mainly due to learning of global representations with transformers. For a given patch, prior work converts spatial information by learning linear combinations of pixels. The global information is then encoded by learning inter-patch information using transformers. As a result, these models lose the image-specific inductive bias, which is inherent to CNNs. As a result, they require more capability to learn visual representations. Hence, they are deep and wide. Unlike these models, our model uses convolution and transformers in such a way that the resulting transformer blocks have convolution-like properties while allowing for global processing. This modeling capability allows us to design shallow and narrow models that are lightweight.

According to the constraints on model parameters, we partition the large-scale search space into three sub-spaces based on parameters and encode them into three independent supernets. Such a partition allows the search algorithm to concentrate on finding models within a specific parameter range, which can be specialized by users according to

their available resources and application requirements. It also reduces gradient conflicts between large and small sub-networks trained via weight-sharing due to gaps in model sizes.

## 3.3 Neural Architecture Search Pipeline

We apply one-shot NAS, which includes two phases: (i) Train a supernet containing all the candidate subnetworks. During training, the parameters of all candidate networks in the search space are optimized simultaneously by weight sharing. (ii) Search for the best sub-network in the well-trained supernet under various resource constraints Cai et al. (2019). Typical search techniques include evolutionary algorithms Guo et al. (2020). Figure 3 shows our overall framework. We use the search space introduced in Table 1 and train a weight-shared supernet by iteratively optimizing randomly sampled subnets from the space. We search for the width, depth, and expansion ratio of both CNN layers and transformer layers and also the number of self-attention heads. The layer and depth in each block are dynamic. After training the supernet, we perform a coarse-to-fine search with hardware latency constraints to find the model with the highest validation accuracy. We propose a latency prediction model, which can efficiently estimate the latency of network candidates by considering network properties, hardware memory access pattern, and degree of parallelism.

### 3.3.1 Supernet Training

In transformer search space, the classical supernet training strategy encounters the following challenges. (i) Slow convergence. This can be attributed to the independent training of transformer blocks resulting in the weights being updated a limited number of times Chen et al. (2021d). (ii) Low performance. The performance of the subnets that inherit weights from the one-shot supernet, trained under classical weight sharing strategy, is far below their performance of training from scratch. This limits the ranking capacity of supernets Anonymous (2022). To mitigate this, existing works perform additional retraining of the searched architectures since their weights are not fully optimized.

Table 1: An illustration of our search space: The search space is divided into three individual supernets within different parameter ranges to satisfy different resource constraints. Tuples of three values in parentheses represent the lowest value, highest, and steps.

| Search Dimension | Width | Depth | Number of Heads | Expansion ratio |
|---|---|---|---|---|
| Supernet XXS | | | | |
| Conv | {16, 24} | - | - | - |
| MBConv-1 | {16, 24} | {1, 2} | - | 1 |
| MBConv-2 | {24, 32} | {2, 3} | - | 1 |
| MBConv-3 | {32, 48} | {2, 3} | - | 1 |
| Transformer-1 | {48, 64} | {2, 3, 4, 5} | {2, 3, 4} | {1.5, 2} |
| Transformer-2 | {64, 80} | {2, 3, 4, 5} | {2, 3, 4} | {1.5, 2} |
| Transformer-3 | {80, 96} | {2, 3, 4, 5} | {2, 3, 4} | {1.5, 2} |
| Transformer-4 | {96, 112} | {2, 3, 4, 5} | {2, 3, 4} | {1.5, 2} |
| MBPool | {1000} | - | - | 6 |
| Number of Stage | | {3, 4} | | |
| Params Range | | $1 \sim 1.8M$ | | |
| Supernet XS | | | | |
| Conv | {16, 24} | - | - | - |
| MBConv-1 | {24, 32} | {1, 2} | - | 1 |
| MBConv-2 | {32, 48} | {2, 3} | - | 1 |
| MBConv-3 | {48, 64} | {2, 3} | - | 1 |
| Transformer-1 | {64, 80} | {2, 3, 4, 5} | {4, 5, 6} | {1.5, 2} |
| Transformer-2 | {80, 96} | {2, 3, 4, 5} | {4, 5, 6} | {1.5, 2} |
| Transformer-3 | {96, 112} | {2, 3, 4, 5} | {4, 5, 6} | {1.5, 2} |
| Transformer-4 | {112,128} | {2, 3, 4, 5} | {4, 5, 6} | {1.5, 2} |
| MBPool | {1000} | - | - | 6 |
| Number of Stage | | {3, 4} | | |
| Params Range | | $1.6 \sim 2.6M$ | | |
| Supernet S | | | | |
| Conv | {16, 24} | - | - | - |
| MBConv-1 | {24, 32} | {1, 2} | - | 1 |
| MBConv-2 | {32, 48} | {2, 3} | - | 1 |
| MBConv-3 | {48, 64} | {2, 3} | - | 1 |
| Transformer-1 | {64, 96} | {2, 3, 4, 5} | {7, 8, 9} | {1.5, 2} |
| Transformer-2 | {96, 128} | {2, 3, 4, 5} | {7, 8, 9} | {1.5, 2} |
| Transformer-3 | {128, 160} | {2, 3, 4, 5} | {7, 8, 9} | {1.5, 2} |
| Transformer-4 | {160, 192} | {2, 3, 4, 5} | {7, 8, 9} | {1.5, 2} |
| MBPool | {1000} | - | - | 6 |
| Number of Stage | | {3, 4} | | |
| Params Range | | $4.0 \sim 7.2M$ | | |

**Weight Entanglement.** Inspired by Autoformer Chen et al. (2021d) operating on transformer-only search space, we apply the weight entanglement training strategy to the vision transformer search space. The central idea is to enable different transformer blocks to share weights for their common parts in each layer. The weight entanglement strategy works specifically for homogeneous building blocks. This is because homogeneous blocks are structurally compatible so that weights can be shared with each other. In the implementation, for each layer, we only need to store the weights of the largest of the $n$ homogeneous candidate blocks. The remaining smaller building blocks can extract the weights directly from the largest building block.

Compared with classical weight-sharing methods, the weight entanglement strategy has three advantages. (i) Faster convergence. Weight entanglement allows each block to be updated more times than the previous independent training strategy. (ii) Low memory cost. We now only need to store the largest building blocks' parameters for each layer, instead of all the candidates in the space. (iii) Better performance of the subnets.

We randomly select a transformer architecture in each iteration. Then we obtain its weights from the weights of the supernet and compute the losses of the subnets. Finally, we update the corresponding weights with the remaining weights frozen. The architecture search space $P$ is encoded in a supernet, denoted as $\mathcal{S}(P, W)$, where $W$ is the weight of the supernet that is shared across all the candidate architectures $s$. Algorithm 1 illustrates the training procedure of our supernet. During the training, the weight $W$ is optimized by:

$$W = \arg \min_{W} \mathcal{L}(\mathcal{S}(P, W)), \qquad (1)$$

where $\mathcal{L}$ represents the loss function on the training dataset. To reduce memory usage, one-shot methods usually sample subnets from $\mathcal{S}$ for optimization. The second stage is to search architectures by ranking the

---

**Algorithm 1** Supernet Training.

---

**Input:** Training epochs $N$ , search space $\mathcal{P}$, supernet $\mathcal{S}$, loss function $L$, train dataset $D_{train}$, initial supernet weights $\mathcal{W}_A$, candidate weights $w$
**for** $i$ in $N$ epochs **do**
    **for** data, labels in $D_{train}$ **do**
        Randomly sample one transformer architecture from search space $\mathcal{P}$
        Obtain the corresponding weights $w$ from $\mathcal{W}_A$
        Compute the gradients based on $L$
        Update the corresponding part of $w$ in $\mathcal{W}_A$ while freezing the rest of the supernet $\mathcal{S}$
    **end for**
**end for**
**Output** $\mathcal{S}$

---

performance of subnets based on the learned weights in $W$. The subnet search objective is:

$$s = \arg \max_{s \in \mathcal{S}} Acc(\mathcal{S}(p, w)), \qquad (2)$$

where $Acc$ indicates the top-1 accuracy of the architecture on the validation dataset, $s$ is the sampled subnet that inherits weight $w$ from $W$.

**Training Efficiency.** We improve training efficiency in two ways. (1) *Search space reduction*: We introduce an inductive bias where the model dimension (width) increases gradually for each transformer stage, resulting in candidates that do not follow the pattern being discarded. This reduces the search space from $10^{16}$ to $10^{10}$ subnets and improves training efficiency. For reference, AutoFormer Chen et al. (2021d) has $10^{16}$ and BigNAS Yu et al. (2020) has more than $10^{12}$ subnets. Similar architectural inductive bias is applied in the stage-based PVT Wang et al. (2021b) and the Swin Transformer Liu et al. (2021). (2) Weight entanglement allows different transformer blocks to share weights of their common parts in each layer with significant benefits of faster convergence, reduced memory footprint, and improved subnet performance.

### 3.3.2 Latency-aware Search Scheme

**Latency Profiling.** Upon acquisition of the trained supernet, we carry out a search algorithm to derive the ideal subnets. Predominant strategies optimize the inference speed of transformers via computational complexity (MACs) or throughput (images/sec) derived from server GPUs. Such metrics, however, fail to accurately reflect real on-device latency. Traditional hardware-aware network search methods usually depend on the hardware deployment of each candidate within the search space to ascertain latency – a process that is both time-consuming and inefficient. A single candidate demands hundreds of inferences to generate an accurate latency, prolonging the search process. Existing works, such as HAT, employ a latency predictor, pre-trained with thousands of real latency data points, as an offline method to forecast the candidate latency, as opposed to obtaining real latency by inference during the search. This method, however, is only applicable to a relatively small search space. For larger search spaces, an increased volume of measured latency data is required as a training set for the predictor, substantially raising the time cost. If the test set is inadequate, the predictor fails to estimate the latency accurately.

To overcome this challenge, we construct a latency lookup table by collecting the on-device latency of MBConv and transformer blocks of varying dimensions. Specifically, the Conv width includes {16,24}, the MBConv width includes {16,24,32,48}, the Transformer block width includes {48,64,80,96,112}, the number of heads includes {2,3,4,5}, and the expansion ratio includes {1.5,2}, thus making a total of 46 modules. It's noteworthy that the speed of individual module execution may differ from when they are combined due to inter-module influences. Given that we do not calculate the latency of each module in isolation, we measure the latency of each module based on the per-latency accuracy drop when removed from the entire model. This methodology provides a more comprehensive and realistic understanding of the latency impacts.

**Search Pipeline.** We propose a coarse-to-fine strategy, which involves integrating latency feedback directly into the design loop when searching for models. This eliminates the need for FLOPs as the latency proxy and reduces the searching time, enabling us to design specialized models efficiently for various hardware platforms.

Figure 3 on the right illustrates our coarse-to-fine strategy. It contains two steps: Initially, we aim to identify a rough skeleton of promising network candidates. Thereafter, we sample multiple fine-grained variations surrounding each skeleton architecture of interest. More specifically, during the coarse-grained phase, we execute a network search using parameters such as memory threshold and perform latency evaluations using the lookup table. Candidates that meet our latency budget are selected to proceed to the fine-grained search phase. Here, we conduct an evolutionary search to procure the optimal subnets (as shown in Figure 5). Our objective is to maximize classification accuracy while minimizing the model size. At the onset of the evolutionary search, we select N random architectures as seeds. The top-k architectures are chosen as parents to generate the next generation via crossover and mutation. For a crossover, two randomly selected candidates are chosen and crossed to produce a new one during each generation. For mutation, a candidate first modifies its depth with a certain probability $P_d$, followed by mutating each block with a probability $P_m$ to generate a new architecture. Algorithm 2 elaborates the procedure of our search method.

---

**Algorithm 2** Latency-aware Coarse-to-fine Search.

---

*Coarse-grained latency guided search:*
**Input:** Latency space $\mathcal{L}$, Search space $\mathcal{P}$, Supernet $\mathcal{S}$, Latency budget $\mathcal{T}_l$, Parameter budget $\mathcal{T}_p$
**for** $i$ in $N$ epochs **do**
    Randomly sample one subnet architecture $\alpha_i$ from $\mathcal{S}$
    Obtain size and latency through $\mathcal{L}$
    **if** Budget of $\alpha_i \leq \mathcal{T}_l \& \mathcal{T}_p$ **then**
        Save subnet to list $\mathcal{A} \leftarrow \{\alpha_i\}$
        $\mathcal{A} :=$ the Top $K$ candidates;
    **end if**
**end for**
**Output** Promising subnet candidates $\mathcal{A}$

*Fine-grained accuracy guided search:*
**Input:** Number of generation iteration $T$, validation dataset $D_{val}$, mutation probability of depth
**Output:** The optimal subnet $\alpha$. $P_d$, mutation probability of each layer $P_m$
**while** search step $t \in (0, T)$ **do**
    **while** $\alpha_i \in \mathcal{A}$ **do**
        Obtain the accuracy of the subnet $N(\alpha_i, W_{\alpha_i})$ on $D_{val}$
    **end while**
    $Top_k :=$ the Top $K$ candidates;
    $Pop_C =$ Crossover$(Top_k, S, C)$
    $Pop_M =$ Mutation$(Top_k, P_d, P_m, S, C)$
    $Pop(t+1) = Pop_C \cup Pop_M$
**end while**

---

Different devices interpret and execute specific operations distinctively due to variations in their underlying architecture, design, computational capabilities, memory management, IO, and bandwidth. Therefore, an operator's latency profile on one hardware platform (e.g., a mobile CPU/GPU) may differ considerably from its profile on another, such as a distinct GPU. This variance stems from the unique characteristics of each hardware. Consequently, it becomes imperative to conduct device-specific optimizations and tests. One cannot simply port results from one hardware setting to another and expect them to remain valid. Nevertheless, crafting multiple lookup tables, although demanding, is substantially more efficient than undertaking multiple comprehensive testing cycles. In our approach, we require data from under 100 instances, while traditional methods might need data from 1000+ instances. Additionally, our method is training-free, whereas conventional techniques demand training, further increasing their resource consumption.

# 4 Experiments

## 4.1 Datasets and Implementation Details

Our experiments are conducted on ImageNet Deng et al. (2009), with approximately 1.2 million images. We report the accuracy on the 50k images in the test set. The image resolution is $256 \times 256$. We train the supernets using a similar recipe as DeiT. Data augmentation techniques, including RandAugment Cubuk et al. (2020), Cutmix Yun et al. (2019), Mixup Zhang et al. (2017) and random erasing Zhong et al. (2020), are adopted with the same hyperparameters as in DeiT except for the repeated augmentation. Images are split into patches of size $16 \times 16$. All the models are implemented using PyTorch 1.7 and trained on Nvidia Tesla V100 GPUs. In the evolutionary search process, we configure a population size of 50 and proceed through 20 generations. Within each generation, we select the top-performing 10 architectures to function as parents, which then produce offspring networks via mutation and crossover mechanisms. We assign mutation probabilities for $P_d$ and $P_m$ as 0.2 and 0.4, respectively.

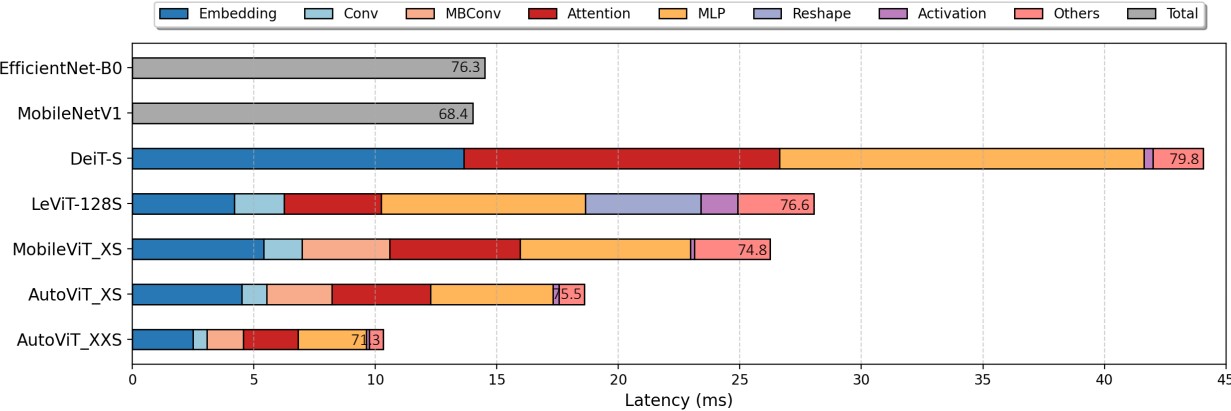

Figure 4: **Latency Breakdown.** Results are obtained on iPhone 13 with CoreML. The on-device speed for various operators is reported. The latency of models and operations are denoted with different colors.

Table 2: Accuracy comparison on ImageNet with state-of-the-art CNN and transformer-based models under similar parameters and computational cost (FLOPs).

| Model | Params (M) | FLOPS (G) | Top-1 Accuracy (%) | Latency (ms) |
|---|---|---|---|---|
| MobileNetv1 | 2.6 | 0.3 | 68.4 | 14.0 |
| MobileNetv2 | 2.6 | 0.2 | 69.8 | 10.7 |
| MobileNetv3 | 2.5 | 0.1 | 67.4 | 8.9 |
| ShuffleNetv2 | 2.3 | 0.1 | 69.4 | - |
| ESPNetv2 | 2.3 | 0.2 | 69.2 | - |
| MobileViT_XXS | 1.3 | 0.2 | 69.0 | 14.7 |
| AutoViT_XXS | 1.8 | 0.3 | 71.3 | **10.2** |
| DeiT-T | 5.7 | 1.3 | 72.2 | 16.7 |
| T2T-T | 4.3 | 1.1 | 71.7 | - |
| PiT-T | 10.6 | 0.7 | 72.4 | 15.4 |
| CrossViT-T | 6.9 | 1.6 | 73.4 | 19.5 |
| EdgeViT-XXS | 4.1 | 0.6 | 74.4 | 28.9 |
| MobileNetV3-L | 5.4 | 0.2 | 75.2 | 26.5 |
| tiny-MOAT-0 | 3.4 | 0.8 | 75.5 | - |
| LeViT-128S | 7.8 | 0.3 | 76.6 | 28.2 |
| MobileViT_XS | 2.3 | 0.6 | 74.8 | 26.5 |
| AutoViT_XS | 2.5 | 0.8 | 75.5 | **19.3** |
| DenseNet-169 | 14.0 | 6.7 | 76.2 | - |
| EfficientNet-B0 | 5.3 | 0.4 | 76.3 | 14.5 |
| AutoFormer-tiny | 5.7 | 1.3 | 74.7 | 25.1 |
| LocalViT-T | 5.9 | 1.3 | 74.8 | - |
| CeiT-T | 6.4 | 1.2 | 76.4 | - |
| PoolFormer-s12 | 12.0 | 2.0 | 77.2 | 59.0 |
| GLIT | 7.2 | 1.4 | 76.3 | - |
| ConvMixer-1024/12 | 14.6 | - | 77.8 | 35.7 |
| NASViT-A0 | 0.3 | 0.2 | 78.2 | - |
| tiny-MOAT-1 | 5.1 | 1.2 | 78.3 | - |
| ResNet50 | 25.5 | 4.1 | 78.5 | 29.4 |
| PVT | 13.1 | 2.1 | 78.7 | 54.5 |
| LeViT-128 | 9.2 | 0.4 | 78.6 | 36.8 |
| MobileViT_S | 5.6 | 1.1 | 78.4 | 35.7 |
| EfficientFormer-L1 | 12.3 | 1.3 | 79.2 | 28.0 |
| DeiT-S | 22.5 | 4.5 | 79.8 | 41.0 |
| AutoViT_S | 6.3 | 1.3 | 79.2 | **27.9** |

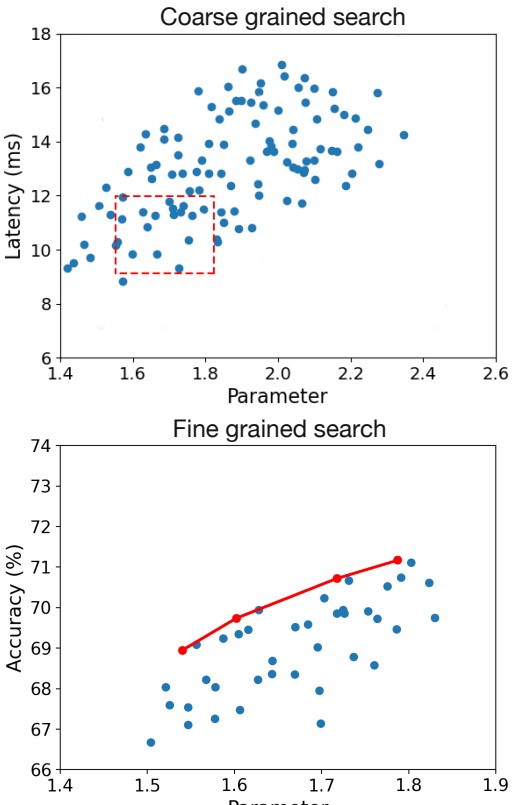

Figure 5: Coarse-to-fine Architecture Search. *Upper figure:* Pre-define some promising candidate network skeletons by the number of parameters and latency budget. *Bottom figure:* Generate more architectures by randomly mutating each skeleton architecture by varying the depth slightly and using the weights from the single-stage model for the induced child models to evaluate all the candidates.

Table 3: Results comparison on COCO2017 object detection and instance segmentation.

| Model | $AP^b$ | $AP^b_{50}$ | $AP^b_{75}$ | $AP^m$ | $AP^m_{50}$ | $AP^m_{75}$ |
|---|---|---|---|---|---|---|
| ResNet18 | 34 | 54 | 36.7 | 31.2 | 51 | 32.7 |
| PoolFormer-S12 | 37.3 | 59.0 | 40.1 | 34.6 | 55.8 | 36.9 |
| EfficientFormer-L1 | 37.9 | 60.3 | 41.0 | 35.4 | 57.3 | 37.3 |
| PVT-Tiny | 36.7 | 59.2 | 39.3 | 35.1 | 56.7 | 37.3 |
| AutoViT_S | 37.6 | 60.2 | 41.1 | 35.3 | 57.5 | 37.6 |

Table 4: Comparison with existing ViT neural architecture search baselines.

| Model | CIFAR10 | CIFAR100 | Flowers | Cars |
|---|---|---|---|---|
| EfficientNet-B0 | 98.1 | 88.1 | 96.9 | 90.8 |
| DeiT-S | 98.0 | 87.1 | 97.8 | - |
| CeiT-T | 98.5 | 88.4 | 96.9 | 90.5 |
| CeiT-T↑384 | 98.5 | 88.0 | 97.8 | 93.0 |
| AutoViT_S | 98.8 | 89.6 | 98.1 | 92.4 |

## 4.2 Experimental Results

We conduct an architecture search on ImageNet and obtain several hybrid transformer models with different parameter sizes. All these models directly inherit the weights from the supernet without additional retraining and other post-processing. We present a summary of performance in Table 2. We compare our hybrid models with various CNN-based and Transformer based models, namely: MobileNets Howard et al. (2017), ShuffleNetv2 Ma et al. (2018), ESPNetv2 Mehta et al. (2019), MobileViT Mehta & Rastegari (2021), DeiT-T Touvron et al. (2021), T2T-T Yuan et al. (2021b), PiT-T Heo et al. (2021), CrossViT-T Chen et al. (2021b), DenseNet-169 Huang et al. (2017), EfficientNet-B Tan & Le (2019), LocalViT-T Li et al. (2021b), CeiT-T Yuan et al. (2021a), PVT Wang et al. (2021b), LeViT Graham et al. (2021), NASViT Anonymous (2022), GLiT Chen et al. (2021a), PoolFormer Yu et al. (2022) and Moat Yang et al. (2022). As shown in Figure 1, our searched hybrid models obtain a better trade-off in model size vs. Top-1 accuracy in ImageNet compared to other models. The configurations of our final searched hybrid models are shown in Table 5. Our AutoViT_XXS, with only 1.8M parameters and 0.3G FLOPs, achieves a Top-1 accuracy of 71.3%, which is higher than all the other CNNs and MobileViT_XXS. Moreover, its latency of 10.2ms is the lowest among the comparable models, thus demonstrating a significant efficiency improvement. Our AutoViT_XS (19.3 ms, 75.5% Top-1 accuracy, 0.8 GFlops) and AutoViT_S (27.9 ms, 79.2% Top-1 accuracy, 1.3 GFlops) also outperform existing ViT variants in both speed and accuracy.

Hand-crafted designs tend to yield a set of models that primarily differ in their dimensions. While they may serve as general-purpose models suitable for a broad range of devices, they may not be as finely optimized for a specific hardware profile as network-searched models. For example, our AutoViT_XS model has 36% fewer parameters (2.5M vs. 3.4M) compared to tiny-MOAT-0m while achieving the same accuracy of 75.5%. This parameter efficiency becomes even more critical when considering the memory constraints typical of edge devices.

We measure the latency of various models on a mobile platform, displayed in Figure 4. Models include MobileNetV1, PiT-T, MobileViT_XS, DeiT-T, and our own model AutoViT_XXS. Different colored bars represent different modules. Notably, our searched model, particularly in the areas of attention and MLP (Multilayer Perceptron), showcases a significant efficiency improvement.

To showcase the generalizability of our method, we evaluate our model on object detection and instance segmentation benchmark, as presented in Table 4. Experiments are conducted on COCO 2017 Lin et al. (2014). We use the Mask R-CNN He et al. (2017) framework and replace different backbones. FLOPs are measured at 1333 × 800 resolution. We also conduct experiments on downstream benchmarks: CIFAR-10 Krizhevsky et al. (2009), CIFAR-100 Krizhevsky et al. (2009), Flowers Nilsback & Zisserman (2008), Cars Krause et al. (2013). Results are presented in Table 5. AutoViT_S demonstrates competitive or superior performance across different tasks and datasets. We achieved an accuracy of 89.6% in CIFAR100, which is 1.2% higher than CeiT-T. We also achieve competitive accuracy compared to EfficientFormer-L1 with fewer parameters (12.3M vs. 6.3M).

## 5 Ablation Study

### 5.1 Knowledge Distillation

In this experiment, we use knowledge distillation to improve the accuracy of the supernet. We use the pre-trained EfficientNet-B5 with 83.3% top-1 accuracy, and RegNetY-32G with 83.6% top-1 accuracy as different teacher models. We also apply soft distillation and hard distillation for comparison. Soft distillation Hinton et al. (2015) minimizes the Kullback-Leibler divergence between the softmax of the teacher and the softmax of the student model. The distillation

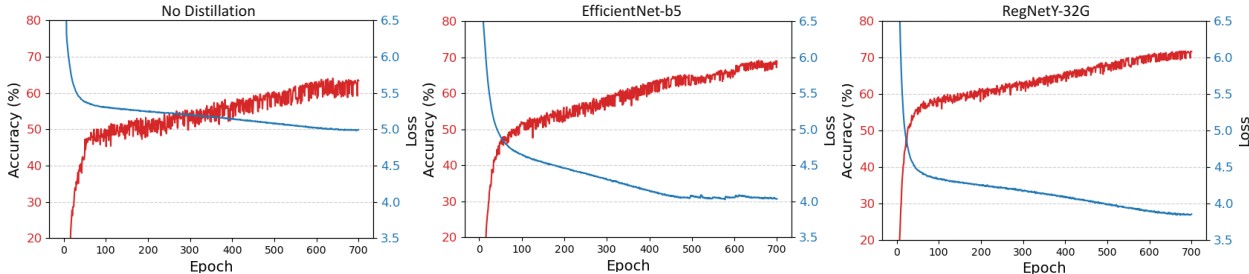

Figure 6: We compare the loss and accuracy convergence curves of the three training methods for Supernet _XXS. From left to right: no distillation, distillation with EfficientNet-B5 as a teacher, and distillation with RegNetY-32G as a teacher. Models trained with RegNetY-32G as the teacher outperform models trained with Efficient-B5 and those without distillation.

Table 5: Our searched hybrid architectures.

| Models | Width | Depth | Number of heads | Expantion ratio |
|---|---|---|---|---|
| AutoViT_XXS | [48, 64, 80, 96, 112] | [5] | [3, 3, 3, 4, 2] | [2, 1.5, 2, 2, 2] |
| AutoViT_XS | [96, 120, 144, 168] | [4] | [4, 6, 4, 4] | [1.5, 2, 2, 2] |
| AutoViT_S | [128, 176, 224, 272] | [4] | [9, 8, 8, 9] | [1.5, 2, 2, 1.5] |

loss is:

$$L_{soft} = (1 - \alpha)L_{CE}(\psi(Z_s), y) + \alpha\tau^2 KL(\psi(\frac{Z_s}{\tau}), \psi(\frac{Z_t}{\tau})), \tag{3}$$

where $Z_t$ and $Z_s$ are the logits of the teacher and student model, respectively. $\psi$ the softmax function. $\tau$ is the temperature for the distillation, $\alpha$ is the coefficient balancing the Kullback–Leibler divergence loss (KL) and the cross-entropy (LCE) on ground truth labels $y$.

For hard-label distillation Touvron et al. (2021), we take the hard decision of the teacher as a true label. The distillation objective is:

$$L_{hard} = (1 - \alpha)L_{CE}(\psi(Z_s), y) + \alpha L_{CE}(\psi(Z_s), y_t), \tag{4}$$

where $y_t = argmax_c Z_t(c)$ is the hard label from the teacher logits $Z_t$. The results are shown in Table 6. When using soft distillation, both EfficientNet-B5 and RegNetY-32G as teachers outperform the model trained without distillation (68.6% vs. 63.5% and 71.3% vs 64.6%). Moreover, we observe that although the accuracy of the RegNetY-32G model is only 0.3% higher than that of the Efficient-B5 model, the accuracy of the supernet model with RegNetY-32G as teacher outperforms the one with Efficient-B5 as a teacher by 2.7%. We also compare hard and soft distillation with RegNetY-32G as a teacher, where hard distillation outperforms soft one by 0.4%. Figure 6 shows the loss and accuracy convergence curves of the three training methods.

## 5.2 Varying Number of Supernets

As discussed in Section 3.2, we partition the large search space into three independent sub-spaces based on parameters and train one supernet for each sub-space. This reduces gradient conflicts between large and small sub-networks trained via weight-sharing due to model size gaps. Further, search space partitioning reduces computational and memory costs. To demonstrate the benefit of our partitioning strategy for supernet training, we perform an experiment with a single supernet covering our hybrid search space. Compared to our XXS model with similar parameters (1.8M), the accuracy of the searched model from the above single supernet achieved a Top-1 accuracy of 65.3%, which is 6.0% lower than the accuracy obtained with the XXS supernet (71.3%).

## 5.3 Hybrid Vs. Transformer-Only Structure.

Table 7 provides a comparison of latency and Top-1 Accuracy between a Transformer-Only model and a Hybrid model. It is observed under similar Top-1 Accuracy (71.2% for Transformer-Only and 71.3% for the Hybrid model), the Hybrid model's latency is significantly lower, approximately 37% less than the Transformer-Only model (10.2ms for the Hybrid

Table 6: Distillation results on ImageNet with Supernet_XXS. We compare the results of our supernet trained with no distillation versus EfficientNet-B5 and RegNetY-32G as teachers for soft and hard distillation.

| Model | Params ($M$) | Top-1 Acc. (%) |
|---|---|---|
| w/o KD | 1.8 | 64.6 |
| Soft KD EffNet | 1.8 | 68.6 |
| Soft KD RegNet | 1.8 | 70.5 |
| Hard KD RegNet | 1.8 | **71.3** |

Table 7: Latency Comparison between Hybrid and Transformer-only Model.

| Model | Latency ($ms$) | Top-1 Acc. (%) |
|---|---|---|
| Transformer-Only | 16.1 | 71.2 |
| Hydbrid | 10.2 | 71.3 |

Table 8: Training time (GPU hours) comparison of different search methods on Supernet_XXS.

| Model | Training Time ($h$) | Top-1 Acc. (%) |
|---|---|---|
| Random | 0 | 66.8 |
| Evolution | 12 | 70.7 |
| Course-to-fine | 5 | 71.3 |

Table 9: Comparison with existing ViT neural architecture search baselines.

| Model | Params($M$) | FLOPS($G$) | Top-1 Acc. (%) |
|---|---|---|---|
| ASViT | 29 | 5.3 | 81.2 |
| GLiT | 7.2 | 1.4 | 76.3 |
| Autoformer | 5.7 | 1.3 | 74.7 |
| AutoViT_S | 6.3 | 1.4 | **79.2** |

model compared to 16.1ms for the Transformer-Only model). This suggests that the Hybrid model can achieve virtually the same performance as the transformer-only model while being significantly more computationally efficient on small lightweight ViT models.

## 5.4 Comparison with Different Search and NAS Methods

**Different Search Methods**. We compare three types of search strategies: random search, evolution search Guo et al. (2020), and our coarse-to-fine search. Evolution search operates in two phases: (i) Crossover: Two randomly selected candidates are crossed to produce a new one. (ii) Mutation. A candidate is chosen randomly and mutated with a probability of 0.1 for each choice to produce a new candidate. Although no training is involved, the process is still time-consuming because the two phases are repeated to generate enough new candidates to obtain those that meet the given architecture constraints. As shown in Table 8, our coarse-to-fine search method can achieve better results (71.3 vs. 70.7 Top-1 accuracy) compared to the previous evolution search with much less training time.

**Different NAS Methods** Table 9 shows a comparison of our method to several ViT-based NAS methods, namely, ASViT Chen (2022), GLiT Chen et al. (2021a), and Autoformer Chen et al. (2021d). Our model offers an excellent balance of efficiency and performance. With only 6.3M parameters and 1.4G FLOPs, AutoViT_S achieves a Top-1 Accuracy of 79.2%. Although slightly lower than ASViT's accuracy (81.2%), AutoViT_S accomplishes this with much less computational demand, emphasizing the model's efficiency. Under similar FLOPs, the Top-1 accuracy of our AutoViT_S is noticeably higher than GLiT (76.3%) and Autoformer (74.7%).

## 6 Conclusion

In this work, we apply neural architecture search to automatically search for optimal lightweight vision transformer (VIT) models given different resource constraints. We design a hybrid search space including CNN and transformer components. We train multiple supernets of different parameter ranges to handle the huge search space in ViT and mitigate conflicts in the weight-sharing of sub-networks. Further, we introduce an efficient latency-aware coarse-to-fine search algorithm to obtain the optimal networks that significantly outperform prior-art lightweight vision transformer models. Our current experiments are limited to classification tasks. Future work includes extending our framework to other vision tasks such as object detection. This work is scientific in nature, we do not believe it has potential negative societal impacts.

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
