# OpenReview forum: "Lightweight Vision Transformer Course-to-Fine Search via Latency Profiling"
_TMLR — Rejected by TMLR_

### Review · Reviewer_sTam · 2023-08-03

**Summary Of Contributions:**

This paper introduces a NAS approach for hybrid deep neural networks (CNN and Transformer). The approach includes three features:
1. This approach can search transformers and CNNs at the same time
2. This approach proposes a coarse-to-fine paradigm to save computation
3. This approach profiles the efficiency using the real latency instead of MACs or FLOPs

**Audience:**

Yes

**Claims And Evidence:**

Yes

**Requested Changes:**

1. There is a typo on page 5, the 3rd line of section 3.3. It should be "(ii)" instead of "(ii))"
2. It would be better if more context on training supernets (one-shot NAS) in the related work.
3. [optional] More benchmarks, e.g. segmentation, detection
4. [optional] Comparison between hand-crafted baseline and NAS hybrid networks (e.g. MOAT)

**Strengths And Weaknesses:**

Strengths

1. The algorithm targets real latency, which makes this algorithm more useful and practical for industrial use.
2. The final models achieve very high accuracy and low latency compared with previous state-of-the-art architecture


Weaknesses:

1. The search space is still limited in certain building blocks. It is unclear whether gains are from the combination of building blocks (MBConv, transformers, standard convs) or the searching algorithm. As a reference, MOAT [1] is also a hybrid architecture combining MBConv and transformers. What is the gap between NAS design and hand-crafted design?

2. Lack of other benchmarks. It would be better to see the detection/segmentation results of the final network for validating the generalization of the networks.

[1] Yang, Chenglin, et al. "Moat: Alternating mobile convolution and attention brings strong vision models." arXiv preprint arXiv:2210.01820 (2022).

---

> ### Author Response · Authors · 2023-10-05
> **Gap between NAS design and hand-crafted design**
>
> The primary benefits of employing Network Architecture Search (NAS) over hand-crafted design are:
> 1. **Efficiency of Design:** NAS reduces the need for manual iterative design processes, allowing for quicker iterations and adjustments based on results.
>
> 2. **Optimal Structures:** An operator's latency on one hardware platform (e.g., a mobile CPU/GPU) may differ considerably from its profile on another, such as a distinct GPU. This variance stems from the underlying architecture, memory management and bandwidth, computation capabilities and processing element (PE) power, reconfigure flexibility, et al. Leveraging network search allows us to pinpoint the optimal structure tailored to specific hardware constraints. Hand-crafted designs tend to yield a set of models that primarily differ in their dimensions. While they may serve as general-purpose models suitable for a broad range of devices, they may not be as finely optimized for a specific hardware profile as network-searched models.  For example, our AutoViT_XS model has 36% fewer parameters (2.5M vs. 3.4M) compared to tiny-MOAT-0m while achieving the same accuracy of 75.5%. This parameter efficiency becomes even more critical when considering the memory constraints typical of edge devices.
>
> The utilization of the latency prediction model is essential, especially when crafting hardware-aware structural designs, which need to consider real hardware features, such as on-device latency, energy consumption, and memory footprint.  Our method is accurate. It considers the properties of the target hardware, the model type, the model size, and the data granularity. Furthermore, it mathematically describes the computation latency and data movement latency to accurately predict the actual throughput of each layer. This model serves as a ready-to-use theoretical framework for general GPU architectures.
>
> We added this discussion in Section 4.2 of the revised paper.

---

> ### Author Response · Authors · 2023-10-05
> **Generalization of the network**
>
> Thank you for highlighting the importance of demonstrating the generalizability of our method. In response to your feedback, we have expanded our benchmarks to object detection and instance segmentation. Results are presented in Table 3 of the revised paper. Experiments are conducted on COCO 2017. We use the Mask R-CNN framework and replace different backbones. FLOPs are measured at 1333 × 800 resolution.
>
> A. Results comparison on COCO2017 object detection and instance segmentation.
> |Models | $AP^b$ | $AP^b_{50}$  | $AP^b_{75}$ | $AP^m$ | $AP^m_{50}$ | $AP^m_{75}$ |
> |------------|-----|-----|-----|-----|-----|-----|
> |ResNet18 |		34 |	54 |	36.7 |	31.2 |	51 |	32.7|
> |PoolFormer-S12  |	37.3	 |59.0 |	40.1 |	34.6 |	55.8 |	36.9|
> |EfficientFormer-L1|	37.9|	60.3|	41.0|	35.4|	57.3|	37.3|
> |PVT-Tiny	|	36.7|	59.2|	39.3|	35.1|	56.7|	37.3|
> |AutoViT\_S   | 37.6 |	60.2|	41.1|	35.3|	57.5|	37.6|

---

> ### Author Response · Authors · 2023-10-05
> **More context on training supernets**
>
> One popular approach to architecture search is the Supernet method. In this method, a single neural network is constructed to represent a large space of possible architectures. During training, the Supernet learns to assign weights to different network parts, effectively selecting a subset of the architecture to be used for a given task. This approach can significantly reduce the computational cost of searching for architectures since the Supernet can be trained once and reused for different tasks. A popular approach is one-shot NAS, where they train one over-parameterized supernet whose weights are shared across all sub-networks in the search space to conduct architecture search, significantly reducing the computational cost. This one-shot NAS approach usually contains two phases. In the initial phase, all candidate networks within the search space are refined using weight sharing, ensuring that every network concurrently achieves optimal performance by the conclusion of training. The subsequent phase employs conventional search methods, like evolutionary algorithms, to identify the top-performing models, considering different resource limitations.
>
> We thank the author for the suggestion. We added this discussion in Section 2  of the revised paper.

---

> ### Author Response · Authors · 2023-10-05
> **Typo**
>
> Thank you for pointing this out. We fixed it in the revised paper.

---

### Review · Reviewer_8Q1B · 2023-08-11

**Summary Of Contributions:**

This paper searches a lightweight hybrid CNN and transformer based neural architectures. The process contains two stages, coarse search stage is latency-awaring and the finegrained search stage is mainly dominated by the validate accuracy. An evolutionary search approach is applied to procure the optimal subnets. In order to mitigate the optimization interference problem, the authors propose a multi-size supernet training scheme.

**Audience:**

Yes

**Broader Impact Concerns:**

No.

**Claims And Evidence:**

Yes

**Requested Changes:**

Please see the weakness. In addition, the readability of this paper can be greatly improved. The writing should be simple and clear, make the article easy to be understanded. The training time of the supernet is not presented and compared.

**Strengths And Weaknesses:**

Strengths:
1. This paper construct a complete neural architecture search framwork. The authors train a supernet firstly and then search the Pareto front by an evolutionary approach.
2. The results of the experiments show that the searched network is superior to some existing methods.

Weaknesses:
1. This paper tends to be an engineering work and the novelty is weak. The supernet pretraining method has been widely used in previous works. The search space and the method of lookup table for latency is not creative.
2. The specific method lacks a detailed description. In Sec 3.2, the alphabetic representation is perplexing and they are not illustrated, such as $X$, $X_{L}$. The used transformer is different from familar transformer which contains $2$ dimensions toekn number an feature dimension, so what is the concrete structure of this transformer?
3. The figures can not express the contents precisely. In Figure.2, what is the meaning of unfold and fold? In Figure.3, what is Attention-FNN?
4. Because the novelty is weak, excellent results are expected. However, in the experiments, the proposed method compares with only part of lightweight methods. There are many lightweight network and NAS based architectures, it is better to compare with some SOTA methods. Besides, as a hybrid method, there are many CNN+transformer architectures in the past two years, the method should compare with them.

---

> ### Author Response · Authors · 2023-10-05
> **Regarding the novelty of the paper**
>
> Thank you for pointing out your concerns regarding the novelty of our work and its potential overlap with existing techniques. Our work synthesizes these components in a novel way to address shortcomings in current research. Previous works have explored these methods to some extent, but they often have with challenges such as:
>
> 1. **Inclusion of Mobile-Unfriendly Operations:** Sophisticated attention mechanisms are hard to support efficiently on mobile devices. The primary reasons stem from the intensive shape manipulations and index operations, which need significant computation cycles and intensive data movement, especially under the limited processing element (PE) power and IO bandwidth of mobile devices. Successful deployment on these platforms necessitates specific algorithm-compilation and algorithm-ComputeLibrary co-design, meaning non-negligible design cost. Operations such as the nonlinear kernel, occupy significant computation cycles. The data layout change can cost GPU/NPU cycles, which computation amounts can not be calculated.
>
> 2. **Reliance on Param/FLOPs as Evaluation Metrics:** Methods like LeViT primarily depend on FLOPs as an efficiency metric, which cannot determine the real on-device speed. Many other factors need to be considered as well, such as the delay of data (movement), and supportability of nonlinear operators.
>
> 3. **Limited Search Spaces:** Most works, like Autoformer, focus only on pure transformers without giving due consideration to latency.
>
> 4. **Extended Training/Search Time:** NASViT's gradient optimization method increases training time for the already high-cost training phase.   HAT's search phase requires extensive data collection and additional predictor training.
>
> In contrast to the aforementioned studies, the primary emphasis of our paper is designing a model with a clear focus on hardware deployment. Our work aims to address the shortcomings of current solutions and to introduce a truly efficient, hardware-friendly approach. This approach is optimized for adapting to hardware limitations and meeting speed requirements. Ensuring that real-world performance aligns more closely with theoretical outcomes. Our core innovations can be distilled as follows:
>
> 1. **Search Space Design:** We incorporated a hybrid search space with an inductive bias. This not only broadens the search space but also accelerates convergence and mitigating local optima. We limit the use of operations that are hard for mobile devices to support.
>
> 2. **Supernet Training:** From a hardware-oriented perspective, given a specific hardware limit, it's more efficient to define a narrowly tailored search space. As a result, there's no need for training with the entire search space. By segmenting the solution into multiple supernets, not only can we sidestep conflicts, but we can also dramatically cut down on training overhead.
>
> 3. **Search Phase:**  Instead of the traditional approach of relying on hardware deployment for every candidate or using a latency predictor, our method stands out in its accuracy and efficiency. Our latency prediction model is a training-free theoretical model, suitable for general-purpose hardware, GPU.  It considers the properties of the target hardware, the model type, the model size, and the data granularity. It then quantitatively captures both the computation latency and data movement latency, enabling it to precisely predict the actual throughput for each layer.
>
> We added this discussion in Section 1 and Section 2 last paragraph of the revised paper.

---

> ### Author Response · Authors · 2023-10-05
> **Detailed description on specific methods**
>
> Thanks for your valuable question, which allows us to clarify our design methodology in detail. We revised the sentence in section 3.2 of the paper. We also revised Figure 2 and Figure 3.
>
> Standard ViT reshapes input tensor X size $H \times W \times C$ into $N \times d$. Where $C, H,$ and $W$ represent the channels, height, and width, N represents the number of patches and d is the dimension. However, reshaping into a 2D feature ignores the spatial inductive bias that is inherent in CNNs.
>
> In Figure 2, the folding-unfolding process replaces local processing in convolutions with global processing using transformers. This allows the transformer block to have CNN- and ViT-like properties, which helps it learn better representations with fewer parameters and simple training recipes.
>
> To learn global representations with spatial inductive bias, \textcolor{blue}{we reshape (unfold) $X$ into $N$ non-overlapping flattened patches $X'$ of size $hw \times N \times d$, where $hw$ is the number of pixels of one patch $P$.  For each patch, inter-patch relationships are encoded by applying transformers for each pixel to obtain.
>
> Simply speaking,  if the original input of the transformer is {$B, N, d$}, with B being the batch size, our input is {$BP, N, d$}, learning more relation between local pixels.  Our transformer structure is the same as the original transformer.
>
>
> In Figure 3, Attention-FNN shows that we search for both attention and FNN (Figure 3 on the left.) of the transformer. It caused some misunderstanding, we changed the  “Attention-FNN” to “Transformer” in the revised Figure 3.

---

> ### Author Response · Authors · 2023-10-05
> **Compare with more SOTA methods**
>
> Thank you for your suggestion. We added more comparisons of recent CNN+transformer architectures in Table 2 of the revised paper.

---

> ### Comment · Reviewer_8Q1B · 2023-10-07
>
> What is the concrete  value of $P$?

---

> > ### Author Response · Authors · 2023-10-07
> > **Value of P**
> >
> > For each patch, the $h$ and $w$ are both 2.  Therefore, our concrete value of $P=hw$ is 4.
> > In our experiment, while a smaller patch size enhances performance, it increases memory usage. Due to GPU constraints, we have set the patch size to 4.

---

### Review · Reviewer_anSj · 2023-09-09

**Summary Of Contributions:**

This work targets latency-aware vision transformer (ViT) neural architecture search (NAS). Specifically, the authors proposed a hybrid search space with both Conv and Transformer in it and built a supernet training scheme on top of it. After supernet training, a latency look-up table is used to find the models with the best accuracy vs. efficiency trade-offs.

**Audience:**

Yes

**Broader Impact Concerns:**

No Broader Impact Concerns

**Claims And Evidence:**

Yes

**Requested Changes:**

I would suggest the authors revise the manuscripts by adding clarification on the "Limited novelty", "Scalability to different devices", and "Limited dataset", as mentioned in the Weaknesses above.

**Strengths And Weaknesses:**

## Strengths
1. Target important question: Although a lot of prior work is compressing ViT models' FLOPs or #params, latency is a more important metric for real applications. Thus, this work explores a unique direction to search ViT with better accuracy vs. latency trade-offs.
2. Informative figures and tables: The paper with well-designed figures and tables is easy to follow.
3. Good performance: the improvement of the accuracy vs. latency trade-offs on ImageNet is obvious.

## Weaknesses
1. Limited novelty: For the claimed contribution of Hybrid Search Space, conv and transformer hybrid structure (in parallel or sequential manner) has been explored in many prior works [1,2,3]. The author did not explain why the newly proposed one is different from prior works clearly. For the claimed contribution of Multi-Size Supernet Training Scheme, it seems to be similar to Autoformer [4]'s weight entanglement, a more detailed comparison between the two will clarify this point in a better way.
2. Scalability to different devices: The authors claimed that "to prevent the need for extensive testing during the search process, we use a lookup table ...". However, if we need to apply the framework to different devices, the "extensive testing" for each device will still be a problem.
3. Limited dataset: The benchmark is only conducted on ImageNet, thus the generalizability to other tasks is unknown.

[1] https://openreview.net/forum?id=Qaw16njk6L

[2] https://arxiv.org/abs/2104.01136

[3] https://arxiv.org/abs/2107.02192

[4] https://arxiv.org/abs/2107.00651

---

> ### Author Response · Authors · 2023-10-05
> **Regarding the novelty and comparison to existing works.**
>
> Thank you for your insightful feedback. We appreciate the opportunity to delineate the distinctions between our proposed approach and existing methods more clearly.  While previous works have showcased improvements in on-device efficiency, they often do not prioritize consideration of the real hardware latency when it comes to designing the DNN models. Our work endeavors to rectify the deficiencies found in existing solutions by introducing a truly efficient, hardware-oriented approach. This approach has been optimized to seamlessly adapt to the constraints of the target hardware and fulfill the specific speed requirements. Below, we contrast the limitations of current methodologies with our proposed solution:
>
> 1.**LeViT** successfully reduces FLOPs at the expense of introducing redundant parameters.
> 1. With the same number of FLOPs,
> LeViT's parameter size is 3-5 times larger than ours. This becomes a significant constraint for edge devices with limited memory capacity.
> 2. Relying solely on FLOPs as a performance metric is misleading since it doesn't necessarily correlate with latency. For instance, despite LeViT-128S having 0.5G fewer FLOPs than our AutoViT_XS, its latency is 7.9ms higher.
>
> 3. The reasons for this discrepancy include: 1) In LeViT, to conduct MHSA, 4D features are frequently reshaped into flat patches. The data layout change can cost GPU/NPU cycles, which computation amounts can not be calculated. 2) The use of HardSwish, which is not inherently supported by iPhone's CoreML, also introduces additional latency overhead. These distinctions are more comprehensively illustrated in the updated Figure 4.
>
> 2.**NASViT** incorporates techniques, such as the gradient projection algorithm and switchable layer scaling design to enhance the convergence and performance of supernet training. However, there are notable drawbacks associated with their approach:
> 1. **Extended Training Time:** The gradient optimization method introduced to circumvent the conflicts between larger and smaller subnets is time-consuming. From a hardware-oriented perspective, it's more efficient to define a narrow search space tailored to specific constraints of target devices. By segmenting the solution into multiple supernets, not only can we sidestep conflicts, but we can also dramatically cut down on training overhead.
> 2. **Search Space** inspired by LeViT, inherits the same drawbacks as the latter.
> 3. **Inclusion of Mobile-Unfriendly Operations**: The 'talking head' module cannot be supported efficiently with a high computational latency.  The nonlinear kernel inside the shifted window attention occupies significant computation cycles. Contrarily, our design minimizes the use of window-based self-attention. This substantial reduction in the number of softmax computations alleviates the negative impact of memory-bound operations on processing speed.
>
> 3.**The Long-Short Transformer** introduces a unique approach by merging long-range attention with dynamic projection to model distant correlations, paired with short-term attention for capturing intricate local details. However, the design presents a set of challenges, particularly when it comes to deployment on mobile platforms:
> 1. **Complex Attention Patterns:** Sophisticated attention mechanisms, like the ones employed in the Long-Short Transformer, are notoriously hard to support efficiently on mobile devices. The primary reasons stem from the intensive shape manipulations and index operations, which need significant computation cycles and intensive data movement, especially under the limited processing element (PE) power and IO bandwidth of mobile devices.
> 2. **Incompatibility with Mobile Devices:** Prevalent mobile devices such as iPhones and Google Pixels do not support the computation design of Long-Short Transformer.
> Successful deployment on these platforms necessitates specific algorithm-compilation and algorithm-ComputeLibrary co-design, meaning non-negligible design cost.
>
> 4.**AutoFormer** is a dedicated one-shot architecture search framework for pure transformer structures. As detailed in Section 3.3. Despite drawing inspiration from AutoFormer for our supernet training method, we have made improvements in various key metrics including training efficiency, accuracy, FLOPs, latency, and parameters:
> 1. **Inductive Bias in Search Space:** Our design incorporates an inductive bias that allows for a gradual increase in model dimension (or width) across each transformer stage. Any candidates that do not adhere to this pattern are promptly eliminated. This strategic reduction narrows down the search space from an overwhelming $10^{16}$ potential subnets to a more manageable $10^{10}$, substantially boosting training efficiency.
> 2. **Latency-Guided Search:** Our latency-focused search methodology empowers the model to identify configurations that are not only smaller and faster but also deliver superior performance compared to what is achievable using AutoFormer.

---

> ### Author Response · Authors · 2023-10-05
> **Scalability to different devices**
>
> We agree with the reviewer. Conducting additional testing for a new device isn't just inevitable but is also a crucial step. Different devices interpret and execute specific operations distinctively due to variations in their underlying architecture, memory management and bandwidth, computation capabilities and processing element (PE) power, reconfigure flexibility, et al.
>
> Therefore, an operator's latency profile on one hardware platform (e.g., a mobile CPU/GPU) may differ considerably from its profile on another, such as a distinct GPU. Consequently, it becomes imperative to conduct device-specific optimizations and tests. One cannot simply port results from one hardware setting to another and expect them to remain valid.
>
> However, our point of emphasis is on the efficiency of the process. Crafting two lookup tables, although demanding, is still substantially more efficient than undertaking two comprehensive testing cycles. In our approach, we require data from under 100 instances, while traditional methods might need data from 1000+ instances. Additionally, our method is training-free, whereas conventional techniques demand training, further increasing their resource consumption.
>
> We added this discussion in 3.3.2 of the revised paper.

---

> ### Author Response · Authors · 2023-10-05
> **Generalizability on different dataset and benchmarks.**
>
> To showcase the generalizability of our method, we evaluate our model on object detection and instance segmentation benchmarks. As presented in Table 3 of the revised paper. Experiments are conducted on COCO 2017. We use the Mask R-CNN framework and replace different backbones. FLOPs are measured at 1333 × 800 resolution.
> We also conduct experiments on downstream benchmarks: CIFAR-10, CIFAR-100, Flowers, and Cars. Results are presented in Table 4 of the revised paper.
>
>
> A. Results comparison on COCO2017 object detection and instance segmentation.
> |Models | $AP^b$ | $AP^b_{50}$  | $AP^b_{75}$ | $AP^m$ | $AP^m_{50}$ | $AP^m_{75}$ |
> |------------|-----|-----|-----|-----|-----|-----|
> |ResNet18 |		34 |	54 |	36.7 |	31.2 |	51 |	32.7|
> |PoolFormer-S12  |	37.3	 |59.0 |	40.1 |	34.6 |	55.8 |	36.9|
> |EfficientFormer-L1|	37.9|	60.3|	41.0|	35.4|	57.3|	37.3|
> |PVT-Tiny	|	36.7|	59.2|	39.3|	35.1|	56.7|	37.3|
> |AutoViT\_S   | 37.6 |	60.2|	41.1|	35.3|	57.5|	37.6|
>
>
> B. Comparison on downstream tasks.
> |Models | CIFAR10|	CIFAR100	|Flowers|	Cars|
> |------------|-----|-----|-----|-----|
> |EfficientNet-B0|	98.1|	88.1|	96.9|	90.8|
> |DeiT-S|	98.0	|87.1|	97.8|	- |
> |CeiT-T        |98.5	|88.4	|96.9|	90.5 |
> |CeiT-T$\uparrow$384       | 98.5	|88.0	|97.8|	93.0 |
> |AutoViT\_S      | 98.8|	89.6|	98.1|	92.4 |

---

### Decision · Action_Editor_nv9v · 2023-12-02

**Recommendation:** Reject

**Comment:**

While the proposed method is reasonable and the reviewers have listed some strengths of this work, the paper in its current form is below the TMLR standard due to several major weaknesses.

1. Insufficient literature review and discussion. The literature survey of this paper is weak and seems to lack enough awareness to the latest development of the area. For example, there is no mentioning of several important efficient ViT backbones, such as FastViT [1], FasterViT [2] and EfficientViT [3]. These methods should be carefully discussed and compared in the experimental section.

2. Novelty. This paper is written in a way as if using lookup table is a novel contribution. As reviewer 8Q1B mentioned, the use of lookup table is not entirely novel and many related works in hardware-latency-aware efficient design could be found with a simple search. It is highly recommended that this part be thoroughly modified and discussed to give enough credits to the community.

3. Experiment setting. The paper's experiment design is weak. Besides missing comparison to the SOTA efficient backbones mentioned above, it is recommended that the authors exactly follow the experimental setting of these papers for apples to apples comparison. In particular, both [2] and [3] have considered TensorRT latency on GPUs. Since the position of this paper emphasizes heavily on hardware-friendly design, it would be great to follow the TensorRT latency measure as well.

[1] FastViT: A Fast Hybrid Vision Transformer using Structural Reparameterization

[2] FasterViT: Fast Vision Transformers with Hierarchical Attention

[3] EfficientViT: Multi-Scale Linear Attention for High-Resolution Dense Prediction

**Audience:**

This work may be interesting to the general efficiency-ware machine learning and edge computing communities. The work may also be interesting to particular areas of applications such as robotics and autonomous driving where efficiency is a critical consideration.

**Claims And Evidence:**

This paper proposes a NAS framework using a lookup table for searching hardware-friendly ViT architectures. The resulting architecture, AutoViT, improves over some architectures such as MobileNetv1/2/3 and MobileViT/PoolFormer on GPU and device (iPhone). The proposed method makes sense from its design. The claims of being state-of-the-art and novel, however, are not well supported due to insufficient literature review and non experimental comparisons. See the comment section for more details.